# *GJB2* and *GJB6* Mutations in Hereditary Recessive Non-Syndromic Hearing Impairment in Cameroon

**DOI:** 10.3390/genes10110844

**Published:** 2019-10-25

**Authors:** Edmond Tingang Wonkam, Emile Chimusa, Jean Jacques Noubiap, Samuel Mawuli Adadey, Jean Valentin F. Fokouo, Ambroise Wonkam

**Affiliations:** 1Department of Pathology, Division of Human Genetics, University of Cape Town, Cape Town 7925, South Africa; wonkamedmond@yahoo.fr (E.T.W.); emile.chimusa@uct.ac.za (E.C.); smadadey@st.ug.edu.gh (S.M.A.); 2Department of Medicine, University of Cape Town, 7925 Cape Town, South Africa; noubiapjj@yahoo.fr; 3Department of Biochemistry, Cell and Molecular Biology, West African Centre for Cell Biology of Infectious Pathogens, College of Basic and Applied Sciences, University of Ghana, Accra 00233, Ghana; 4Department of Surgery, ENT unit, Bertoua Regional Hospital, P.O. Box 40 Bertoua, Cameroon; valentin.fokouo@gmail.com

**Keywords:** hearing impairment, genetics, *GJB2* and *GJB6*, Cameroon, Africa

## Abstract

This study aimed to investigate *GJB2* (connexin 26) and *GJB6* (connexin 30) mutations associated with familial non-syndromic childhood hearing impairment (HI) in Cameroon. We selected only families segregating HI, with at least two affected individuals and with strong evidence of non-environmental causes. DNA was extracted from peripheral blood, and the entire coding region of *GJB2* was interrogated using Sanger sequencing. Multiplex PCR and Sanger sequencing were used to analyze the prevalence of the *GJB6*-D3S1830 deletion. A total of 93 patients, belonging to 41 families, were included in the analysis. Hearing impairment was sensorineural in 51 out of 54 (94.4%) patients. Pedigree analysis suggested autosomal recessive inheritance in 85.4% (35/41) of families. Hearing impairment was inherited in an autosomal dominant and mitochondrial mode in 12.2% (5/41) and 2.4% (1/41) of families, respectively. Most HI participants were non-syndromic (92.5%; 86/93). Four patients from two families presented with type 2 Waardenburg syndrome, and three cases of type 2 Usher syndrome were identified in one family. No *GJB2* mutations were found in any of the 29 families with non-syndromic HI. Additionally, the *GJB6*-D3S1830 deletion was not identified in any of the HI patients. This study confirms that mutations in the *GJB2* gene and the del(*GJB6*-D13S1830) mutation do not contribute to familial HI in Cameroon.

## 1. Introduction

Hearing impairment (HI) is a disabling congenital disease with the highest rate for age-standardized disability of life in the world [1]. Globally, congenital HI has a prevalence of 1.3 per 1000 population [2], and is accounted for in about 1 per 1000 live births in developed countries, with a much higher incidence of up to 6 per 1000 live births in sub-Saharan Africa [3]. Genetic factors contribute from 30 to 50% of hearing impairment cases in sub-Saharan Africa [4]. In 70% of neonates who fail newborn hearing screens (NBHS) and are presumed to have inherited HI, there are no other distinguishing physical findings and the HI is classified as non-syndromic [5]. Among non-syndromic (NS) HI, nearly 80% of cases are inherited in an autosomal recessive (AR) mode [6,7].

Non-syndromic hearing impairment (NSHI) is an extremely heterogeneous trait, with approximately 170 NSHI loci and 112 genes identified to date [8]. Nevertheless, studies in European and Asian populations have identified pathogenic variants in *GJB2* (encoding connexin 26) and *GJB6* (encoding connexin 30) as the major contributors to autosomal recessive NSHI (ARNSHI) [9], with *GJB2*-c.35delG being the most prevalent variant (20–50%) found in cases of autosomal recessive non-syndromic hearing impairment (ARNSHI) [7,10]. The *GJB6*-D13S1830 deletion was identified in up to 9.7% of cases, and thus is the second largest contributor to the genetic etiology of NSHI in European populations, either with homozygous presentation, or when present in addition to a *GJB2* mutation on the opposite allele [11,12].

Cameroon is a sub-Saharan African country, covering an area of 475,442 km^2^, with a 2017 census reporting a population of 24,053,727 [13]. Cameroon is frequently referred to as “Africa in miniature”, because of its many geographical and cultural attributes, its population and linguistic diversity (there are more than 200 distinct local languages in the country) [14], and its vast genetic diversity that mimics that of Africa [15].

Previous studies have found no contribution of the *GJB2* and *GJB6* genes to HI in Cameroon [16,17,18]. However, the patients included in those studies were chosen indiscriminately, and consisted of both familial and isolated cases, with a high likelihood of environmental causes in many cases. In this study we revisit the contribution of the *GJB2* and *GJB6* genes to HI in Cameroon by only focusing investigations into cases that showed a clear pattern of inheritance within families.

## 2. Materials and Methods

### 2.1. Ethical Approval

The study was performed in accordance with the Declaration of Helsinki. Ethical approval was obtained from the Institutional Research Ethics Committee for Human Health of the Gyneco-Obstetric and Pediatric Hospital of Yaoundé, Cameroon (No. 723/CIERSH/DM/2018), and the University of Cape Town’s Faculty of Health Sciences’ Human Research Ethics Committee (HREC 104/2018). Written and signed informed consent was obtained from all participants who were 21 years of age or older, and from parents or guardians in cases of minors, with verbal assent from participants, including permission to publish photographs.

### 2.2. Participant Selection

Hearing impaired patients were recruited from eight of the ten administrative regions of Cameroon, from schools for the deaf, and in the community, following procedures previously reported in Cameroon [14]. Briefly, all participants’ detailed personal and family histories were obtained, medical records were reviewed by a general practitioner, a medical geneticist, and an ENT specialist when possible, and relevant data were extracted, including three-generation pedigrees and perinatal histories. A general systemic and otological examination and audiological evaluation were performed, including pure tone audiometry. We followed the recommendation number 02/1 of the Bureau International d’Audiophonologie (BIAP), Belgium, to classify hearing levels [14,19]. Only HI individuals belonging to families with segregating hearing impairment, with at least two affected individuals, and with strong evidence of non-environmental causes were recruited.

### 2.3. Molecular Analysis

Genomic DNA samples were extracted from peripheral blood, following the manufacturer’s instructions for the available commercial kit (Puregene Blood Kit^®^ (Qiagen, Alameda, CA, USA)), at the Biochemical Laboratory of the Centre Pasteur du Cameroun, Yaoundé, Cameroon, or using the Chemagic extraction protocol, in the division of Human Genetics, University of Cape Town, South Africa.

Previously reported primers for the *GJB2* gene were evaluated using BLAST® (NIH, USA) and other software, as recommended [17]. The entire coding region of the *GJB2* gene (exon 2) was amplified, followed by sequencing using an ABI 3130XL Genetic Analyzer® (Applied Biosystems, Foster City, CA, USA), in the Division of Human Genetics, University of Cape Town, South Africa.

Detection of del(*GJB6*-D13S1830) was performed using the method and primers described by del Castillo et al. [11,12]. The entire coding region of *GJB6* was amplified using the method described by Chen et al. [20]. The PCR results were validated by Sanger sequencing of 10% of the samples.

### 2.4. Data Analysis

Data analysis was performed through the use of descriptive statistics.

## 3. Results

### 3.1. Participant Demographics

We recruited a total of 93 patients belonging to 41 families. Their mean age was 18 ± 10.4 (1–50) years. The male/female ratio was 0.82 (42/51). Hearing impairment was congenital in 62 patients (66.7%), and the mean age at medical diagnosis was 3.2 ± 3.3 (1–22) years.

### 3.2. Audiometric Characteristics

Pure tone audiometry was performed in 54 of our 93 patients. Hearing impairment was sensorineural in 51 out of 54 (94.4%) patients and mixed in 3 patients; no patients exhibited a conductive hearing impairment. All of our patients had bilateral hearing impairment, and the majority had profound to total hearing impairment (n = 51; 94.4%) (Table 1).

### 3.3. Inheritance Pattern

Pedigree analysis suggestive of autosomal recessive inheritance was the most frequently observed pattern of inheritance and accounted for 85.4% (35/41) of families (Figure 1A). In 12.2% (5/41) and 2.4% (1/41) of families, HI were likely inherited in an autosomal dominant and mitochondrial mode, respectively (Figure 1B). Consanguinity was present in three families (3/41; 7.3%). The inbreeding coefficient was 0.0625 in one of these families (Figure 1A), and 0.0156 in the other two families. A total of six participants (6/93; 6.5%) were thus born from consanguineous union.

### 3.4. Non-Syndromic and Syndromic Hearing Impairment

The majority of our participants exhibited non-syndromic hearing impairment, which accounted for 92.5% (86/93) of cases, for a total of 38 families (Figure 2).

Four patients from two families in our cohort presented type 2 Waardenburg syndrome (without dystopia canthorum). The main clinical signs included: hearing impairment, patches of hypopigmented skin, sapphire-blue eyes, and premature white hair (Figure 3).

Three cases of type 2 Usher syndrome (without vestibular areflexia), from one family, were identified; in addition to hearing impairment, clinical signs of retinitis pigmentosa were present, including night vision impairment and constricted visual field.

### 3.5. Molecular Analysis Results of GJB2 and GJB6

A total of 29 families with segregating recessive non-syndromic hearing impairment were tested for mutations in *GJB2* and for the del(*GJB6*-D13S1830) mutation. None of the families exhibited the del(*GJB6*-D13S1830) mutation, or any of the reported disease-causing mutations in *GJB2.* However, a *GJB2* variant of uncertain significance, NM_004004.6: c.499G>A (p.V167M), was present in one family in the heterozygous form (Appendix A).

## 4. Discussion

This report is the most compressive study of the role of *GJB2* and *GJB6* in familial HI in Cameroon, and confirmed the non-implication of these genes in non-syndromic HI in that country; this is consistent with previous reports in selected isolated HI cases of putative genetic origin [16,17]. By carefully and stringently selecting only multiplex families in the current studies, our results consolidate previous findings. In addition, we recruited in nearly all the schools, as well as in the community, for the deaf in 8/10 provinces representing about 90% of the population; therefore, we are confident that the sample is representative of the population in Cameroon.

The low implication of the *GJB2* gene in non-syndromic hearing impairment has also been demonstrated in other populations of African descent. The 35delG mutation, which constitutes almost 50% of all *GJB2* mutations in Caucasians [21,22,23], was not reported in 406, 356, 182, and 126 probands from Kenya, Ghana, South Africa, and Uganda, respectively [24,25,26,27]. Moreover, no pathogenic mutations in *GJB2* were found in a cohort of 44 probands from Nigeria with non-syndromic deafness [28], and only 7% of a cohort of 127 American probands of Hispanic or African descent with bilateral non-syndromic hearing impairment presented a disease causing mutation in *GJB2* [29]. However, an African exception is the Ghanaian population, where the *GJB2* founder mutation p.R143W (c.427C˃T) was shown to be highly prevalent in that population [30], and accounted for nearly 25% of familial cases and 8% of isolated cases of HI [31]. The *GJB2* variant, c.499G>A (p.V167M), of uncertain significance according to the ClinVar database, was present in a family in our cohort, and has previously been described by our group in the Cameroonian population [17]. It has also been described in the USA [32] and in China [33]. The *GJB2* variant c.499G>A (p.V167M) could thus be considered to require further investigation.

The most common mutation in *GJB6* is a 342-kb deletion, *GJB6*-D13S1830, which causes non-syndromic HI when homozygous, or when present on the opposite allele to a *GJB2* mutation [24,34]. The del(*GJB6*-D13S1830) mutation is the second most frequent genetic cause of non-syndromic prelingual hearing impairment in the Spanish population (after the 35delG mutation in *GJB2*) [11]. This deletion is also prevalent in France, Brazil, and Israel [12,35,36], but is rare or absent in Italy (i.e., Sicily), Romania, Iran, and India [7,37,38,39,40,41]. This deletion was also absent in Nigerians [28] and in Ghanaians [31]. In order to identify any other mutations different from del(*GJB6*-D13S1830), the coding region of *GJB6* was sequenced in African probands from Cameroon, South Africa, and Uganda; however, this revealed no additional pathogenic mutations [18,26].

Our findings support previous reports that *GJB2* and *GJB6* do not play a significant role in non-syndromic hearing impairment in most populations of African descent. Interestingly, genetic testing through targeted genomic enrichment and massively parallel sequencing of 116 genes were used to screen 10 multiplex families with non-syndromic hearing impairment. In 7 of the 10 families (70%), 12 pathogenic variants were identified in 6 genes, and nearly half of these variants were novel [42]. Therefore, due to the highly heterogeneous genetic nature of NSHI, next generation sequencing would be the most effective way to identify variants associated with non-syndromic deafness in the African populations [4], and all the families investigated in the present study should be subjected to whole exome sequencing in order to potentially identify variants in known genes as well as novel genes. Indeed, based on the identification of specific inner ear transcripts, it is estimated that more than 1000 NSHI genes are still to be identified [43].

The study also confirms Waardenburg syndrome as the most common cause of syndromic hearing impairment in Cameroon, in line with previous reporting in other African populations [44]; these families, as well as the singular family displaying mitochondrial inheritance, should also warrant further molecular investigation.

## 5. Conclusions

The present study showed that hereditary hearing impairment in Cameroon is mostly non-syndromic, congenital, sensorineural, and inherited in an autosomal recessive mode. Additionally, this study identified Waardenburg and Usher syndromes as the most common syndromic hearing impairments in Cameroon. This study confirmed that mutations in the *GJB2* gene and the del(*GJB6*-D13S1830) deletion are not implicated in familial non-syndromic hearing impairment in Cameroon. Future studies should employ whole genome sequencing approaches and functional genomics studies to identify other genes that may be implicated in the hearing impairment observed in these families.

## Figures and Tables

**Figure 1 genes-10-00844-f001:**
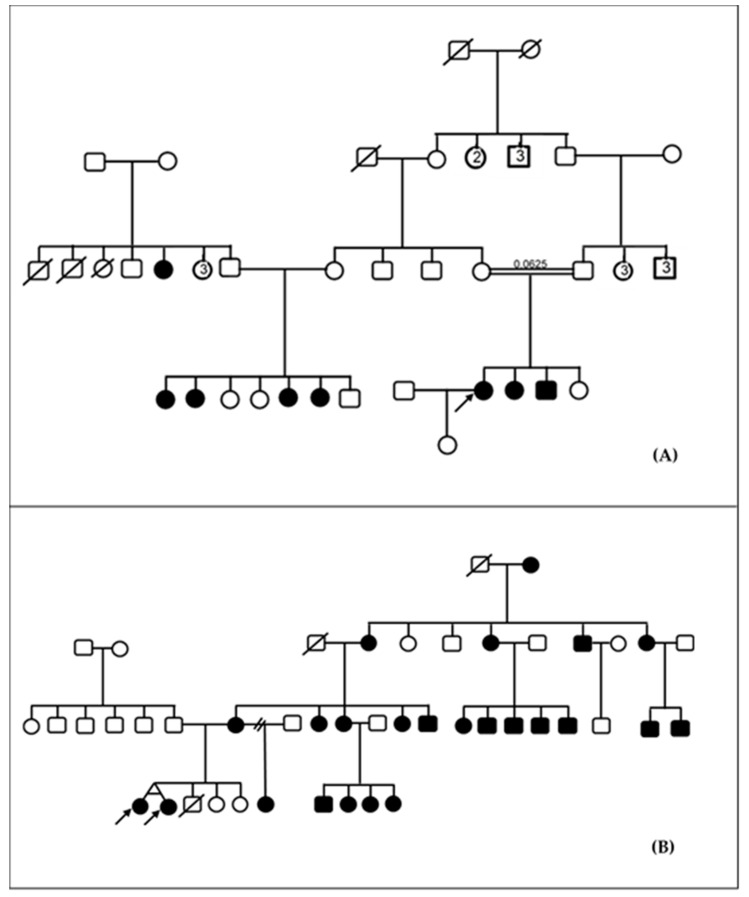
Inheritance of familial hearing impairment in Cameroon. (**A**) Pedigree of a consanguineous family with autosomal recessive non-syndromic hearing impairment. (**B**) Pedigree of a family with non-syndromic hearing impairment suggestive of mitochondrial inheritance. Arrows here indicate the probands.

**Figure 2 genes-10-00844-f002:**
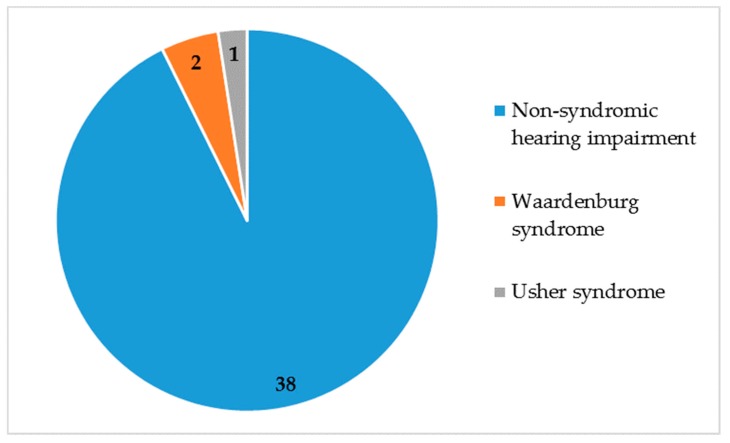
Non-syndromic and syndromic hearing impairment. N = 41 families.

**Figure 3 genes-10-00844-f003:**
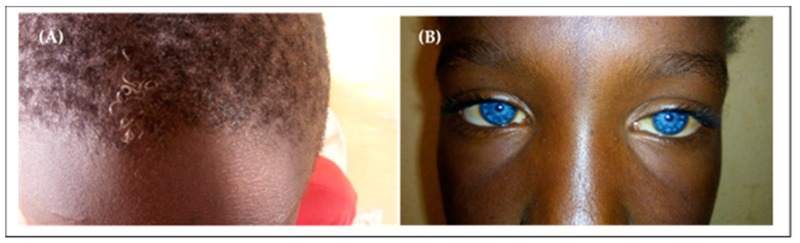
Waardenburg syndrome in our cohort. (**A**) Premature white hair; (**B**) Sapphire-blue eyes.

**Table 1 genes-10-00844-t001:** Audiometric classification of hearing impairment, according to the Bureau International d’Audiophonologie (BIAP) recommendation.

Level of Hearing Loss	N *	Percentage (%)
Severe I (71–80 dB)	01	1.8
Severe II (81–90 dB)	02	3.7
Profound I (91–100 dB)	04	7.4
Profound II (101–110 dB)	13	24.1
Profound III (111–119 dB)	23	42.6
Total (≥120 dB)	11	20.4
Total	54	100

* Number of patients.

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
