# Peer review of "GJB2 and GJB6 Mutations in Hereditary Recessive Non-Syndromic Hearing Impairment in Cameroon"

_genes, 2019, doi:10.3390/genes10110844_

Round 1

Reviewer 1 Report

the paper builds on previous studies of the authors, and as such is an addition to the growing knowledge that the DFNB1 locus is not a major cause of hearing loss in African populations. The paper is nicely written and results are clearly presented, but I have my doubts about the originality of the paper.

minor remarks:

when mentioning a sequence variant, please include the reference sequence used to annnotate the variant (in this case probably NM_004004.5 of NM_004004.6)

Fig 1b: in this case it was not molecularly proven that it really mitochondrial inheritance, hence it would be better to refer to the pedigree data as suggestive of mitochondrial inheritance

when referring to an overview on GJB2/GJB6 (DFNB1) please use a more recent paper than ref 10

Author Response

Reviewer 1

Reviewer’s 1 comment 1: The paper builds on previous studies of the authors, and as such is an addition to the growing knowledge that the DFNB1 locus is not a major cause of hearing loss in African populations. The paper is nicely written, and results are clearly presented, but I have my doubts about the originality of the paper.

Authors’ response 1: We thank the reviewer for the positive appreciation of our work. We do recognize previous works on GJB2 and GJB6 mutations in Hearing impairment in Cameroon. Nevertheless, we will like to stress that by carefully and stringently selecting only multiplex families in the current studies, our results we consolidate previous findings.

minor remarks:

Reviewer’s 1 comment 2: When mentioning a sequence variant, please include the reference sequence used to annotate the variant (in this case probably NM_004004.5 of NM_004004.6);

Authors’ response 2: The amendments have been added, as follows: The section Results/ Molecular Analysis Result of GJB2 and GJB6/P5/L139-140 now read: “…However, a GJB2 variant of uncertain significance, NM_004004.6:c.499G>A (p.V167M), was present in one family in the heterozygous form…”

Reviewer’s 1 comment 3: Fig 1b: in this case it was not molecularly proven that it really mitochondrial inheritance, hence it would be better to refer to the pedigree data as suggestive of mitochondrial inheritance

Authors’ response 3: The change has been implemented, as follows: Pedigree of a family with non-syndromic hearing impairment suggestive of mitochondrial inheritance.

In addition, the section Results/Inheritance pattern/P3/L109-112 now reads: Pedigree analysis suggestive of autosomal recessive inheritance was the most frequently observed pattern of inheritance and accounted for 85.4% (35/41) of families (Figure 1A). In 12.2% (5/41) and 2.4% (1/41) of families, HI were  likely inherited in an autosomal dominant and mitochondrial mode, respectively (Figure 1B).

Reviewer’s 1 comment 4: when referring to an overview on GJB2/GJB6 (DFNB1) please use a more recent paper than ref 10

Authors’ response 4 : We have added a more recent paper as follows

Del Castillo, F.J.; Del Castillo, I. DFNB1 Non-syndromic Hearing Impairment: Diversity of Mutations and Associated Phenotypes. Front Mol Neurosci 2017, 10:428.

Reviewer 2 Report

The article entitled "GJB2 and GJB6 mutations in Hereditary Non-syndromic Hearing Impairment in Cameroon" by Wonkam et al reports on genetic screening of major known deafness-causing mutations in European/Asian populations (GJB2/GJB6) in Cameroonian families segregating hearing loss.
Major points. As the authors acknowledge there is already literature supporting no main contribution of these genes in Cameroon and Sub-Saharian Africa (Trotta 2011 and Bosch 2014). In this perspective the present article brings no much novelty. What attracts more attention is that most families have recessive inheritance suggesting strong genetic factors in the population. I recommend to emphasize this aspect of the work starting from rephrasing the title. The authors should provide data showing what is the prevalence of recessively inherited deafness over the total interviewed patients/families and reassuring that the sample is representative of the whole population of Cameroon. The authors should detail on consanguinity vs. non-consanguinity in families and should show the pedigree-based inbreeding coefficient. It would be interesting to know the regional distribution of the ascertained families to elaborate on possible founder effects. Minor points. Is is not clear whether GJB6 deletion only or the whole gene was screened. "Italy AND Sicily" should be corrected as Sicily is part of Italy while it seems as a separate country here.

Author Response

Reviewers  2

Reviewer’s 2 comment 1. The article entitled "GJB2 and GJB6 mutations in Hereditary Non-syndromic Hearing Impairment in Cameroon" by Wonkam et al reports on genetic screening of major known deafness-causing mutations in European/Asian populations (GJB2/GJB6) in Cameroonian families segregating hearing loss.
Major points. As the authors acknowledge there is already literature supporting no main contribution of these genes in Cameroon and Sub-Saharian Africa (Trotta 2011 and Bosch 2014). In this perspective the present article brings no much novelty.

Authors’ response 1.  We thank the reviewer for the positive appreciation of our work. We do recognize previous works on GJB2 and GJB6 mutations in Hearing impairment in Cameroon. Nevertheless, we will like to stress that by carefully and stringently selecting only multiplex families in the current studies, our results we consolidate previous findings.

We have added the later point in the discussion.

Reviewer’s 2 comment 2. What attracts more attention is that most families have recessive inheritance suggesting strong genetic factors in the population. I recommend to emphasize this aspect of the work starting from rephrasing the title.

Authors’ response 2.  Thanks for the suggestions: we have amended the title as follows: “GJB2 and GJB6 mutations in Hereditary Recessive Non-Syndromic Hearing Impairment in Cameroon”

In addition, the section Results/Molecular analysis of GJB2 and GJB6/P5/L136-137 now reads: A total of 29 families with segregating recessive non-syndromic hearing impairment were tested for mutations in GJB2 and for the del(GJB6- D13S1830) mutation.

Reviewer’s 2 comment 3. The authors should provide data showing what is the prevalence of recessively inherited deafness over the total interviewed patients/families and reassuring that the sample is representative of the whole population of Cameroon.

Authors’ response 3. As stated, in the results section, we recruited a total of 93 patients, belonging to 41 families. Autosomal recessive inheritance was the most frequently observed pattern of inheritance and accounted for 85.4% (35/41) of families

We recruited in nearly all the school for the deaf in 8/10 province representing nearly 90% of the populations, as well as in the community (Figure S2). We are therefore confident the sample is representative of the population in Cameroon

The point above has been added in the discussion. Thanks!

Reviewer’s 2 comment 4. The authors should detail on consanguinity vs. non-consanguinity in families and should show the pedigree-based inbreeding coefficient.

Authors’ response 4. We provided more details on consanguinity, and the section Results/Inheritance pattern/P3/L114-116 now reads: “Pedigree-based consanguinity was present in three families (3/41; 7.3%). The inbreeding coefficient was 0.0625 in one of these families (Figure 1A), and 0.0156 in the other two families. A total of six participants (6/93; 6.5%) were thus born from consanguineous union.”

Reviewer’s 2 comment 5. It would be interesting to know the regional distribution of the ascertained families to elaborate on possible founder effects.

Authors’ response 5. The regional distribution of the ascertained families is now displayed on the supplementary File S2.

Minor points.

Reviewer’s 2 comment 6. It Is is not clear whether GJB6 deletion only or the whole gene was screened.

Authors’ response 6. As stated in the method, we only investigated GJB6-D3S1830 deletion

Reviewer’s 2 comment 7. "Italy AND Sicily" should be corrected as Sicily is part of Italy while it seems as a separate country here.

Authors’ response 7. The correction is made.

Round 2

Reviewer 2 Report

I have no further comments